# Highly Active and Carbon-Resistant Nickel Single-Atom Catalysts for Methane Dry Reforming

**Mohcin Akri [1], Achraf El Kasmi [2], Catherine Batiot-Dupeyrat [3],\* and Botao Qiao [1,4],\***

1. CAS Key Laboratory of Science and Technology on Applied Catalysis, Dalian Institute of Chemical Physics, Chinese Academy Sciences, Dalian 116023, China; akri.mohcin@dicp.ac.cn
2. Institute of Engineering Thermophysics, Chinese Academy of Sciences, Beijing 100190, China; achraf@iet.cn
3. Institut de Chimie des Milieux et Matériaux de Poitiers (IC2MP), Ecole Nationale Supérieure d'Ingénieurs de Poitiers (ENSIP), Université de Poitiers, UMR CNRS 7285, 1rue Marcel Doré, TSA 41105, CEDEX 9, 86073 Poitiers, France
4. Dalian National Laboratory for Clean Energy, Dalian Institute of Chemical Physics, Chinese Academy of Sciences, 457 Zhongshan Road, Dalian 116023, China
\* Correspondence: catherine.batiot.dupeyrat@univ-poitiers.fr (C.B.-D.); bqiao@dicp.ac.cn (B.Q.)

**Abstract:** The conversion of $CH_4$ and $CO_2$ to syngas using low-cost nickel catalysts has attracted considerable interest in the clean energy and environment field. Nickel nanoparticles catalysts suffer from serious deactivation due mainly to carbon deposition. Here, we report a facile synthesis of Ni single-atom and nanoparticle catalysts dispersed on hydroxyapatite (HAP) support using the strong electrostatic adsorption (SEA) method. Ni single-atom catalysts exhibit excellent resistance to carbon deposition and high atom efficiency with the highest reaction rate of 1186.2 and 816.5 $mol.g_{Ni}^{-1}.h^{-1}$ for $CO_2$ and $CH_4$, respectively. Although Ni single-atom catalysts aggregate quickly to large particles, the polyvinylpyrrolidone (PVP)-assisted synthesis exhibited a significant improvement of Ni single-atom stability. Characterizations of spent catalysts revealed that carbon deposition is more favorable over nickel nanoparticles. Interestingly, it was found that, separately, $CH_4$ decomposition on nickel nanoparticle catalysts and subsequent gasification of deposit carbon with $CO_2$ resulted in CO generation, which indicates that carbon is reacting as an intermediate species during reaction. Accordingly, the approach used in this work for the design and control of Ni single-atom and nanoparticles-based catalysts, for dry reforming of methane (DRM), paves the way towards the development of stable noble metals-free catalysts.

**Keywords:** nickel single atom catalysts; atoms efficiency; nanoparticles; dry reforming; hydroxyapatite

## 1. Introduction

Dry reforming of methane (DRM) to generate synthesis gas is gaining more and more attention as a potential renewable system in the energy sector. The reaction converts the two main greenhouse gases [1–3] into a mixture of hydrogen and carbon monoxide with a low $H_2/CO$ ratio suitable for methanol and Fischer–Tropsch synthesis [4,5]. The reaction is strongly endothermic ($\Delta H_{298K}$ = + 247 kJ/mol) and requires a high reaction temperature (>700 °C) in the presence of supported catalysts to activate $CH_4$ and $CO_2$ [6]. DRM has been reported extensively over numerous catalysts based on noble metals, such as, Ir, Rh, Ru, Pt, and Pd [7,8], and transition metals, especially Co and Ni [9–13]. Although noble metal-based catalysts are relatively more resistant to the undesirable coking compared to non-noble metal catalysts [14], a large-scale commercialization is limited by their high cost and limited availability. On the contrary, transition metal catalysts are much cheaper and more

available but tend to deactivate more quickly than noble metals due to carbon deposition (coke). The two reactions involved in carbon deposition are: $CH_4$ decomposition ($CH_4 \leftrightarrow C + 2H_2$) and Boudouard reaction ($2\,CO \leftrightarrow CO_2 + C$) [15,16].

It has been reported that the size of supported nickel particles is a critical parameter for the DRM performance in terms of intrinsic activity, stability, and resistance to carbon deposition [17–19]. The use of a very small particle size or, more interestingly, atomically dispersed active phase, is highly desired to maximize the atom efficiency and avoid the accumulation of carbon [20–22]. However, small nanoparticles tend to agglomerate rapidly to larger particles at high temperature, causing catalyst deactivation [23]. Hence, efforts were focused on the design of nickel catalysts with high resistance to sintering phenomena. Recently, the synergy of single-atom sites $Ni_1$ and $Ru_1$ anchored on the $CeO_2$ surface of $Ce_{0.95}Ni_{0.025}Ru_{0.025}O_2$ was evidenced by Tang et al. The authors showed that atoms remain singly dispersed and in a cationic state during catalysis up to 600 °C [24].

Moreover, the preparation of atomically dispersed metals catalysts has recently emerged as a promising method over various supports [22,25–30]. Thus, the required proprieties of the catalytic support could be summarized as follows: (i) high affinity of metals with surface groups of the support materials, (ii) high density and dispersion of metal, and (iii) strong metal support interaction (SMSI).

The promising behavior of hydroxyapatite ($Ca_{10}(PO_4)_6(OH)_2$: HAP) supported Ni catalysts in the DRM reaction was first demonstrated by Boukha et al. [31]. Moreover, SMSI between Au nanoparticles and HAP was evidenced [32]. According to many published papers, varying the Ca/P molar ratio leads to the formation of additional phases and induces significant changes in the textural and acid-base properties [33]. The presence of both cations ($Ca^{2+}$ and $Ca^{1+}$) and anions ($PO_4^{3-}$ and/or $OH^-$) can also be considered as important sites for anchoring and stabilizing metals single atoms [34,35].

Herein, we report a comparative study of nickel single-atom and nanoparticles deposited on hydroxyapatite support (Ni/HAP) for DRM reaction. Two different methods of synthesis were used, namely a classical one named strong electrostatic adsorption (SEA) and a second one using the surfactant polyvinyl pyrrolidone (PVP) possessing hydrophilic (pyrrolidone) and hydrophobic groups (polyvinyl backbone). The catalysts were characterized before and after reaction using numerous advanced techniques.

## 2. Experimental Section

### 2.1. Catalysts Preparation

#### 2.1.1. Synthesis of Hydroxyapatite (HAP) Support

A chemical co-precipitation method using ammonium dihydrogen phosphate (($NH_4)_2\,HPO_4$) and calcium nitrates ($Ca(NO_3)_2$, $4H_2O$) was applied to synthesize stoichiometric hydroxyapatite (Ca/P = 1.67) [32]. Thus, the proper amounts of both salts are dissolved separately in 80 mL of deionized water and the pH was adjusted to 10.5 by the addition of ammonia 25% solution. Then, ammonium dihydrogen phosphate solution was added (1 mL.min$^{-1}$) dropwise to the calcium nitrates. The resultant milky suspension was stirred at 90 °C with a speed of 600 rpm for 2 h, and the precipitate was rinsed with deionized water several times before being filtered prior to drying at 80 °C overnight. The dried sample was then calcined at 400 °C for 4 h in static air and the resulting powder was sized to 50–100 μm.

#### 2.1.2. Deposition of Ni on HAP

Small loading (0.5 wt%) and high loading (5 and 10 wt%) of Ni were deposited on hydroxyapatite support to obtain single-atom and nanoparticle catalysts, respectively, by a simple method named strong electrostatic adsorption (SEA) at room temperature [36]. Typically, an adequate amount of nickel nitrates ($Ni(NO_3)_2$, $6H_2O$) was dissolved in deionized water (100 mL) followed by the adjustment of pH to 10 using ammonia solution. Afterwards, 1 g of HAP was added to the prepared solution and

stirred for 24 h at room temperature. For PVP-stabilized nickel single-atoms, an adequate amount of nickel and PVP (50 mg) were dissolved separately in the deionized water and added together to the support (HAP), the obtained precipitate was filtered, rinsed carefully with deionized water, and finally dried at 80 °C overnight before being calcined at 500 °C for 4 h. The obtained samples were donated as xNi/HAP where x represents the weight loading of Ni. For instance, the 0.5 wt% Ni single-atom catalyst is denoted as $0.5Ni_1/HAP-SAC$ while 5 and 10wt% Ni/HAP are named as 5Ni/HAP and 10Ni/HAP, respectively, and the $0.5Ni_1/HAP-SAC$ catalyst assisted-PVP named as $0.5Ni_1/HAP-SAC-PVP$.

## 2.2. Catalysts Characterization

### 2.2.1. Inductively Coupled Plasma Spectrometry-Atomic Emission Spectrometry (ICP-AES)

Ni content in all prepared samples was determined by inductively coupled plasma spectrometry-atomic emission spectrometry (ICP-AES) with an IRIS Intrepid II XSP instrument (Thermo Electron Corporation).

### 2.2.2. Specific Surface Areas (SSA)

Textural properties were measured by $N_2$ adsorption-desorption isotherms recorded at 77 K using a Micromeritics ASAP 2460-4 apparatus. Specific surface areas were determined from the adsorption branch using the Brunauer–Emmett–Teller method, whereas pore diameters determined from the desorption branch by using the BJH method.

### 2.2.3. X-ray Diffraction (XRD)

XRD patterns were recorded with a PW3040/60 X'Pert PRO PANalytical diffractometer with Cu $K_\alpha$ radiation (0.15432 nm) operating at 40 kV and 40 mA. Patterns were recorded in the range of $2\theta = 10–80°$.

### 2.2.4. Temperature-Programmed Reduction ($H_2$-TPR)

Temperature-programmed reduction ($H_2$-TPR) was performed using Micromeritics AutoChem II 2920 instrument. Subsequently, a quantity of 150 mg of each sample was placed in a quartz reactor and pretreated in flowing Ar (30 mL $min^{-1}$) at 573 K for 30 min and then decreased to room temperature. After purge for 30 min, the temperature was increased to 1073 K with a ramp of 10 °C $min^{-1}$ under a flow of 10% $H_2/Ar$ (50 mL $min^{-1}$) and $H_2$ consumption detected by TCD.

### 2.2.5. Scanning Transmission Electron Microscopy (STEM)

High-angle annular dark-field scanning transmission electron microscopy (HAADF-STEM) examination was performed on JEOL JEM-2100F instrument while aberration-corrected HAADF-STEM was performed on a JEOL JEM-ARM200F apparatus equipped with a CEOS probe corrector, which provides a resolution of 0.08 nm. Before measurement the samples were ultrasonically dispersed in ethanol and dropped onto carbon films.

### 2.2.6. X-ray Photoelectron Spectroscopy (XPS)

Chemical state of Ni was examined by XPS which was performed on a Thermo Fisher ESCALAB 250Xi instrument equipped with a monochromated $AlK\alpha$ source ($h\nu = 1486.6$ eV, 15 kV and 10.8 mA).

### 2.2.7. Raman Spectroscopy

Raman spectra were collected with a LabRam HR800 confocal microprobe Raman instrument (HORIBA Jobin Yvon) with a laser power of ca. 0.1 mW and laser excitation at 532 nm (He–Ne laser).

### 2.2.8. X-ray Absorption Fine Structure (XAFS)

The X-ray absorption exanimation was performed in National Synchrotron Radiation Research Center at the bending magnet beamline BL12B of SPring-8 (8 GeV, 100 mA), where the X-ray monochromatic beam was focused with the beam size of 0.5 mm in the vertical direction and 2.0 mm in the horizontal direction around sample position. The crystal structures of Ni foil and NiO are taken from Materials Project (https://materialsproject.org).

### 2.3. Catalytic Test

The catalytic performance was evaluated at atmospheric pressure using a fixed bed quartz reactor with equimolar amounts of $CH_4$ and $CO_2$ ($CH_4$:$CO_2$: He = 1:1:3) with a total flow rate of 50 mL/min at 750 °C. The catalyst amounts of 50, 6.16, and 3.65 mg of catalysts for $0.5Ni_1$/HAP-SAC, 5Ni/HAP-NP, and 10Ni/HAP-NP, respectively, were used to maintain the same metal amount (see Figure S1), generating a metallic gas hourly space velocity (MGHSV) of $12 \times 10^6$ mL $g_{Ni}^{-1}$ $h^{-1}$. Before each catalytic test, the samples were in situ reduced at 500 °C for 60 min under 10 vol% $H_2$/He flow, and then purged for 30 min with pure He. The reactor temperature was recorded with a thermocouple inserted into the catalytic bed of the reactor. The reactants and products were analyzed by Agilent 6890 gas chromatograph (GC) equipped with a thermal conductivity detector and a TDX-01 column connected online with the setup.

For the separately decomposition of $CH_4$, the experiments were carried out in the same setup using the similar amount of catalyst (50 mg) by increasing temperature (10 °C.min$^{-1}$) from room temperature up to 800 °C under pure $CH_4$ flow. After cooling down to room temperature under helium gas, 10 mL min$^{-1}$ of pure $CO_2$ was employed to gasify the previously deposited carbon to CO. The temperature ramped from room temperature to 800 °C with a rate of 25 °C.min$^{-1}$. The reactants and products were analyzed online by using a mass spectrometer. The $CO_2$ and $CH_4$ conversions were determined according to the following formula:

$$X_{CH4} = \frac{F_{CH_4,in} - F_{CH_4,out}}{F_{CH_4,in}} \times 100$$

$$X_{CO2} = \frac{F_{CO_2,in} - F_{CO_2,out}}{F_{CO_2,in}} \times 100$$

where $F_{i,in}$ and $F_{i,out}$ are the inlet and outlet gas flow rates of i gas, respectively. Methane and $CO_2$ reaction rate (mol.$g_{Ni}^{-1}$.h$^{-1}$) were calculated from:

$$Rate\ CH_4 = \frac{(F_t * [CH_4] * X_{CH_4})}{(Weight\ of\ catalyst \times w_{Ni} \times 22.4 \times 10^3)} \times 60$$

$$Rate\ CO_2 = \frac{(F_t * [CO_2] * X_{CO_2})}{(Weight\ of\ catalyst \times w_{Ni} \times 22.4 \times 10^3)} \times 60$$

where $F_t$ represents the total flow (mL.min$^{-1}$) and $[CO_2]$, $[CH_4]$, $X_{CH_4}$, and $X_{CO_2}$ denote the concentrations (vol%) and conversions of $CO_2$ and $CH_4$, respectively, whereas $w_{Ni}$ is the nickel metal loading (wt%). Turnover frequencies (TOF s$^{-1}$) defined as the number of $CH_4$ or $CO_2$ molecule converted per catalytic site per second, this parameter is determined based on the following equation:

$$TOF = \frac{(F_t * [CH_4] * X_{CH_4})}{Weight\ of\ catalyst \times W_{Ni} \times \frac{Dispersion}{58.7} \times 22.4 \times 10^3 \times 60}$$

## 3. Results and Discussions

### *3.1. Textural Proprieties*

Supplementary Figure S2 shows the $N_2$ physical adsorption–desorption isothermal curve of hydroxyapatite support (HAP) after 4 h of pretreatment under static air at 400 °C. The curve exhibits the hysteresis loop of type-IV isotherm curves [37]. The average pore diameter was found to be 14.8 nm, the surface area is 109 $m^2$/g, and the pore volume is 0.46 $cm^2$/g (micropore volume = 0.006 $cm^2$/g).

### *3.2. Structural Properties*

XRD patterns of the as-prepared HAP support and xNi/HAP catalysts with different Ni loading (x = 0.5, 5 and 10 wt%) are displayed in Figure 1a. The diffraction patterns show a unique crystalline phase corresponding to stoichiometric hydroxyapatite $Ca_{10}(OH)_2(PO_4)_6$ (ICDD 00-001-1008). Moreover, no signals evidencing the existence of NiO or $Ni(OH)_2$ have been detected even at a high loading of nickel (10 wt%).

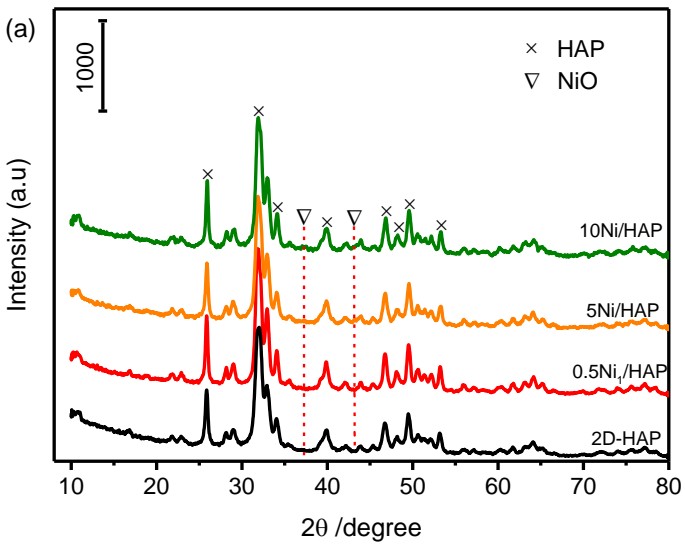

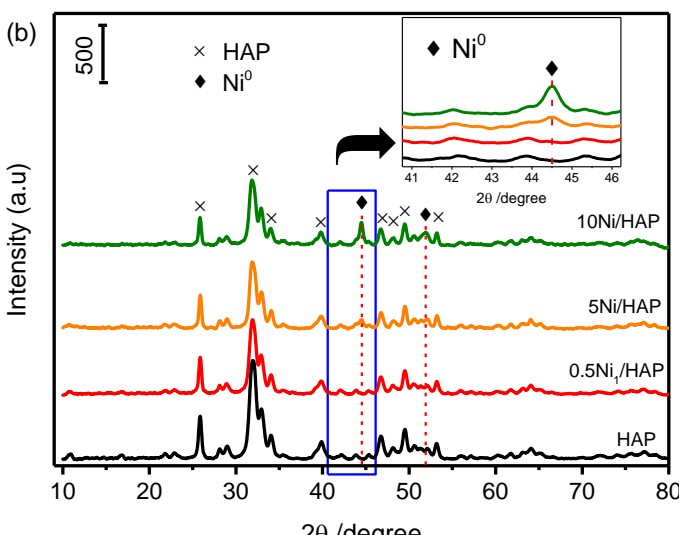

**Figure 1.** XRD patterns of hydroxyapatite (HAP) and Ni/HAP, before reduction (**a**), after reduction at 500 °C (**b**).

Furthermore, since the catalysts were reduced by $H_2$ at 500 °C before catalytic tests, XRD measurements were performed after the reduction step. As shown in Figure 1b, the catalyst with low-loading of Ni ($0.5Ni_1$/HAP-SAC) exhibits no Ni peaks. On the other hand, a peak corresponding to metallic Ni was detected at high loading (5 and 10%Ni/HAP-NP). Nickel was not detected at low loading, which indicates either a high dispersion of Ni on HAP support or the formation of very small Ni nanoparticles on the external surface of HAP that are below the detection limit of XRD. In order to examine the possibility of having low nickel loading at the surface of catalysts, we analyzed its chemical composition using X-ray photoelectron spectroscopy.

### 3.3. Chemical States of Ni over Various Catalysts

To determine the surface chemical composition of the catalysts surface and the chemical state of nickel single atoms, X-ray photoelectron spectroscopy (XPS) analysis was performed for the reduced samples at 500 °C. The survey spectra recorded for the reduced catalysts with different nickel loadings (Figure 2a) exhibited the presence of nickel even at low Ni loading and both Ca and P elements.

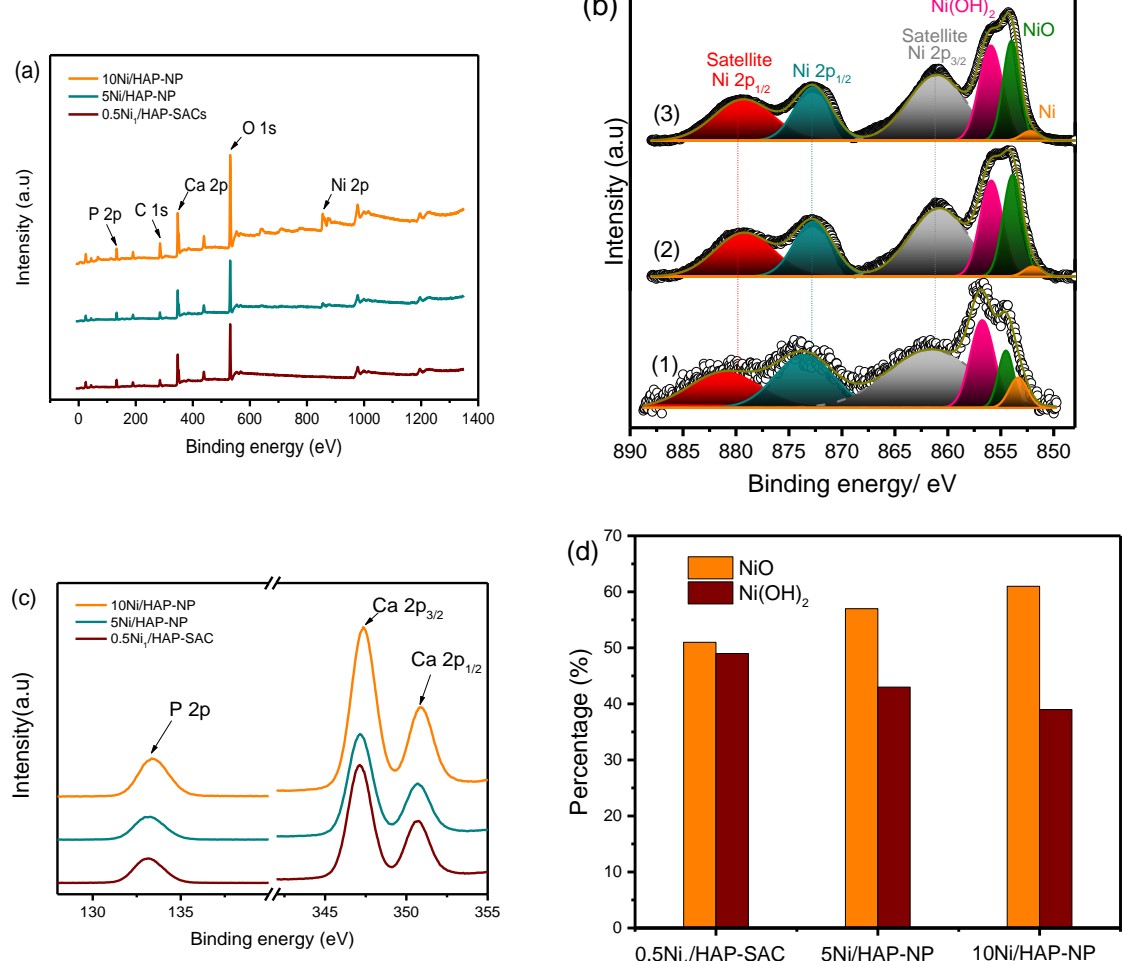

**Figure 2.** (**a**) The survey XP spectra of Ni/HAP catalysts; (**b**) High resolution Ni 2p XP spectra of Ex-situ reduced (**1**) $0.5Ni_1$/HAP-SAC, (**2**) 5Ni/HAP-NP, and (**3**) 10Ni/HAP-NP; (**c**) High resolution P 2p and Ca 2p XP spectra of ex situ reduced samples; (**d**) NiO and $Ni(OH)_2$ percentage as a function of Ni loading.

The Ni 2p spectrum (Figure 2b) exhibits complex features associated to the multiplet splitting of the Ni 2p at 852.6, 853.7 and 855.8 eV, assigned respectively to metallic Ni, NiO and $Ni(OH)_2$ [22]. The satellite structure at 861.2 eV characterizes $Ni^{2+}$ likely stabilized as NiO. However, the weak peak intensity observed related to $Ni^0$ is probably due to the re-oxidation of nickel after the exposition of

samples to the atmospheric oxygen. It is interesting to note that a correlation was observed between nickel loading and the molar ratio $NiO/Ni(OH)_2$ (Figure 2d). An increase of the nickel loading is accompanied by an increase in $NiO/Ni(OH)_2$ ratio.

H$_2$-TPR profiles (Figure 3) show three reduction peaks centered at about 350, 430, and 500 °C. The intensity of the first peak increases significantly with nickel loading, mostly due to the reduction of surface NiO species in weak interaction with HAP. The second peak can be attributed to the reduction of well dispersed NiO particles in strong interaction with HAP and the third one could be attributed to Ni species incorporated into the HAP structure, as proposed by Boukha et al. [33].

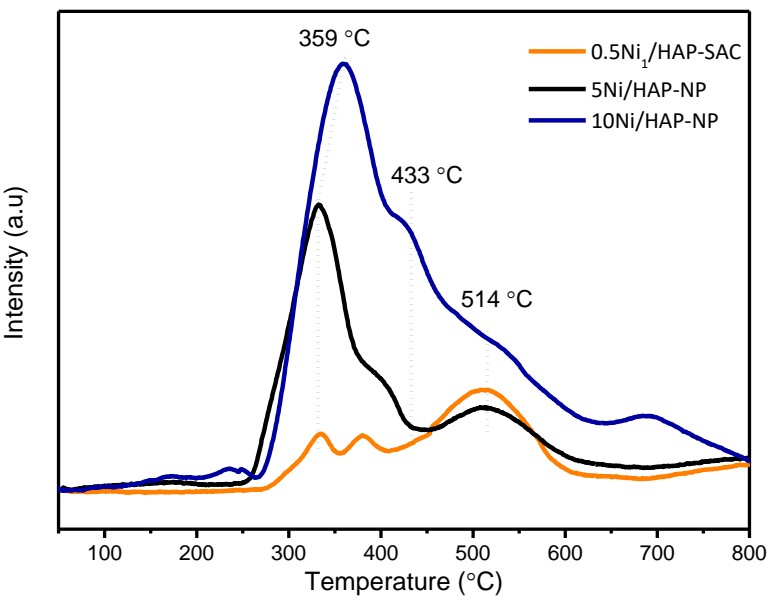

**Figure 3.** TPR-H$_2$ diagrams of x%Ni/HAP catalysts with different nickel loading amount (x = 0.5; 5; and 10 wt.% Ni).

### 3.4. Morphology and Microstructure

The STEM images of 5Ni/HAP-NP and 10Ni/HAP-NP show the presence of high density of nickel nanoclusters with average particle size ≤1 nm (Supplementary Figure S3). After hydrogen reduction pretreatment at 500 °C, a slight increase in the average particle size is observed for 5Ni/HAP-NP (Figure 4a) and 10Ni/HAP-NP (Figure 4b) with 2.1 nm and 2.6 nm, respectively. More STEM and HRTEM images for 10Ni/HAP-NP are presented in Supplementary Figure S4 and Figure 4c, respectively. Additionally, a small fraction (<5%) of large nanoparticles ~10 nm were observed only for 10Ni/HAP-NP as indicated by STEM image (Supplementary Figure S4b). In addition, elemental mapping of 5Ni/HAP revealed a uniform distribution of Ni on the surface of HAP with the absence of large nanoparticles, as shown in Supplementary Figure S5. For low Ni loading, the as-prepared 0.5Ni$_1$/HAP-SAC catalyst, STEM images show the absence of nanoparticles (Figure 5a). Moreover, the examination with ac-HAADF-STEM highlighted the presence of individual and uniformly isolated nickel single atoms (Figure 5b and Supplementary Figure S6). Figure 5c represents the extended X-ray absorption fine structure spectra (EXAFS) of 0.5Ni$_1$/HAP-SAC after in-situ reduction at 500 °C and the reference spectra of Ni foil and NiO at the Ni L2-edge using a Fourier transform. One prominent peak is at ~1.5 Å from the Ni–O contribution and a relatively weak peak at ~2.3 Å probably stemmed from the Ni–Ni contribution, confirming the presence of predominant Ni single atoms and the relatively small percentage of Ni sub-nanoclusters in the 0.5Ni$_1$/HAP catalyst after reduction at 500 °C. Corresponding Ni X-ray absorption near-edge spectra (XANES) collected after in-situ reduction at 500 °C showed that the white-line intensities in the spectra of 0.5Ni$_1$/HAP-SAC is close to that of NiO (Figure 5d), implying that Ni single atoms in the catalyst remain in a high oxidation state. Finally, we could reason

out that low-Ni loading allows the design of atomically dispersed Ni catalysts (e.g., $0.5N_1$/HAP-SAC), whereas at high-Ni loading, Ni nanoparticles are obtained (e.g., 5Ni/HAP-NP).

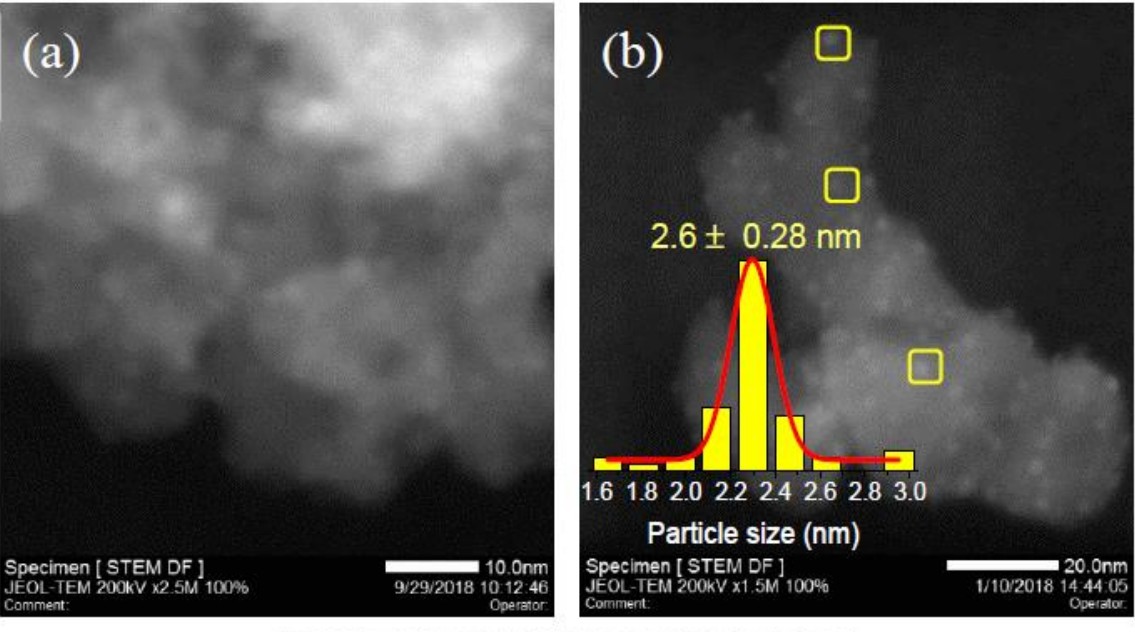

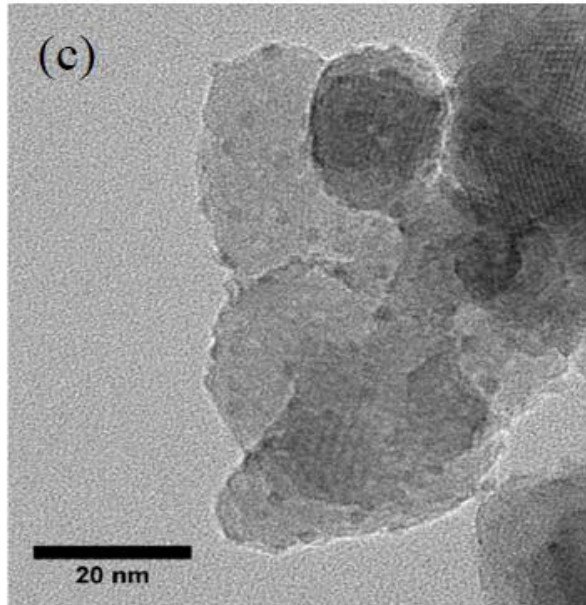

**Figure 4.** STEM images of (**a**) 5Ni/HAP-NP, (**b**) 10Ni/HAP-NP catalysts reduced at 500 °C, (**c**) HRTEM image of 10Ni/HAP-NP catalyst reduced at 500 °C yellow squares indicate respectively some typical nickel nanoparticles.

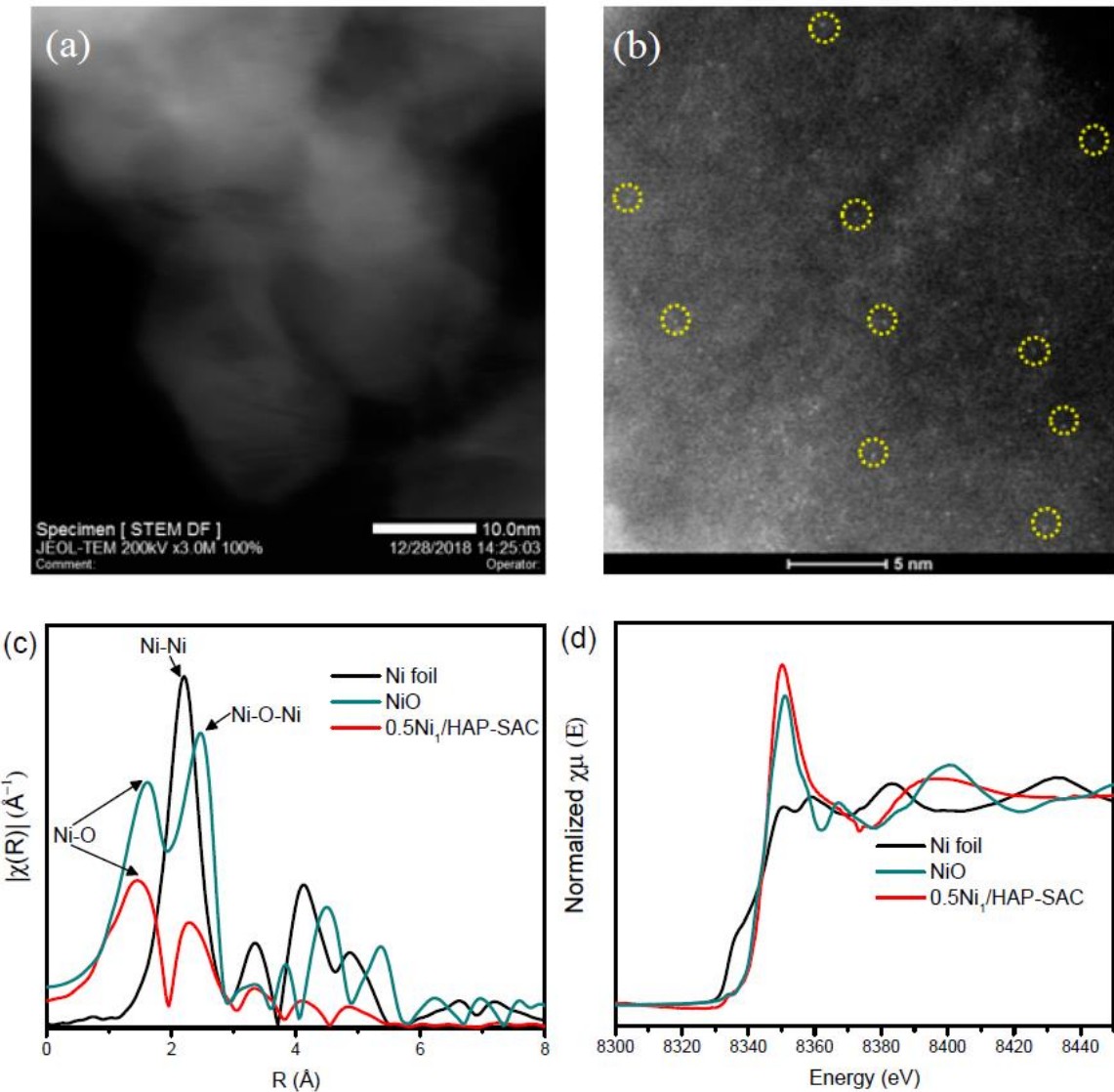

**Figure 5.** (**a**) STEM image of as-prepared $0.5Ni_1$/HAP-SAC catalyst; (**b**) Aberration-Corrected scanning transmission electron microscopy (AC-STEM) for as-prepared $0.5Ni_1$/HAP-SAC catalyst; (**c**) Fourier transform EXAFS spectrum of the in-situ reduced $0.5Ni_1$/HAP-SAC catalyst at 500 °C in comparison with NiO and Ni foil at the Ni $L_2$-edge; (**d**) XANES spectra at the Ni $L_2$-edge of NiO, Ni foil and reduced $0.5Ni_1$/HAP-SAC catalyst at 500 °C.

### 3.5. Catalysts Performance

3.5.1. Influence of Ni Loading

DRM is a highly endothermic reaction that often requires high temperature to activate both $CH_4$ and $CO_2$ molecules [15]. In order to identify the temperature range in which the activation takes place over nickel single-atom and nanoparticle catalysts, both $0.5Ni_1$/HAP-SAC as single-atom catalyst and 5Ni/HAP-NP as nanoparticle catalyst were exposed to the mixture of $CH_4$ and $CO_2$ using a continuous fixed bed flow reactor, under atmospheric pressure, in a stepwise heating mode from room temperature up to 800 °C. Figure 6 shows the mass spectrometer (MS) signals of the reactants $CH_4$ ($m/z$ = 16), $CO_2$ ($m/z$ = 44) and products ($H_2$: $m/z$ = 2, CO: $m/z$ = 28) over $0.5Ni_1$/HAP-SAC and 5Ni/HAP-NP catalysts. It is observed that, at temperatures up to around 450 °C, the concentrations of the reactants and the products remained practically constant. At temperatures higher than 450 °C and 400 °C for $0.5Ni_1$/HAP-SAC and 5Ni/HAP-NP, respectively, the concentrations of both $CH_4$ and $CO_2$ started to

decrease progressively and syngas ($CO+H_2$) appeared in the gas phase, indicating the onset of the methane dry reforming reaction. A further increase of temperature resulted in an increase of CO and $H_2$ concentration up to reach the maximum at 800 °C.

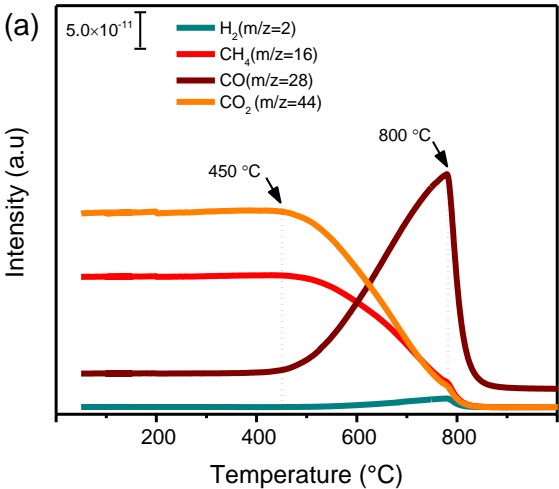

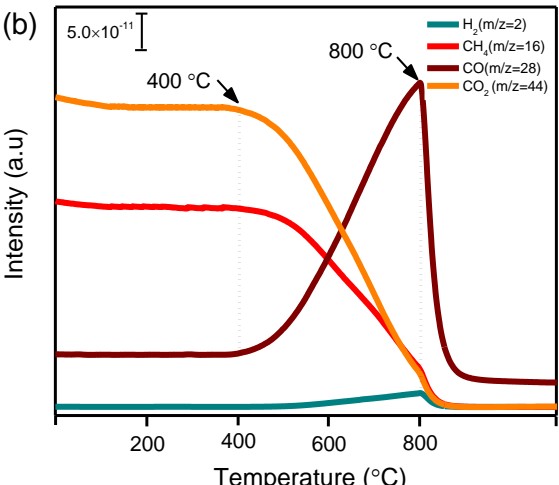

**Figure 6.** Mass spectrometer (MS) signal of $CH_4$, $CO_2$, $H_2$ and CO during dry reforming of methane over (**a**) nickel single-atoms catalyst (0.5Ni$_1$/HAP-SAC) and (**b**) nanoparticles catalyst (5Ni/HAP-NP).

The stability DRM reaction over nickel single-atom and nanoparticle catalysts was studied at 750 °C, using 50, 6.16, and 3.65 mg of catalysts for 0.5Ni$_1$/HAP-SAC, 5Ni/HAP-NP and 10Ni/HAP-NP, respectively, which corresponds to similar metallic gas hourly space velocity. As shown in Figure 7, initially, $CH_4$ and $CO_2$ conversions are the same whatever the catalyst composition. It can be seen that $CH_4$ conversion decreased by 25%, 26%, and 36% over 0.5Ni$_1$/HAP-SAC, 5Ni/HAP-NP, and 10Ni/HAP-NP, respectively, after 630 min of reaction. The $CO_2$ conversion followed the same trend and remained higher than the $CH_4$ conversion because of the occurrence of a reverse water gas shift reaction (Equation (1)).

$$CO_2 + H_2 \rightleftharpoons CO + H_2O \quad \Delta H_{298K} = +41.2 \text{ kJ.mol}^{-1} \tag{1}$$

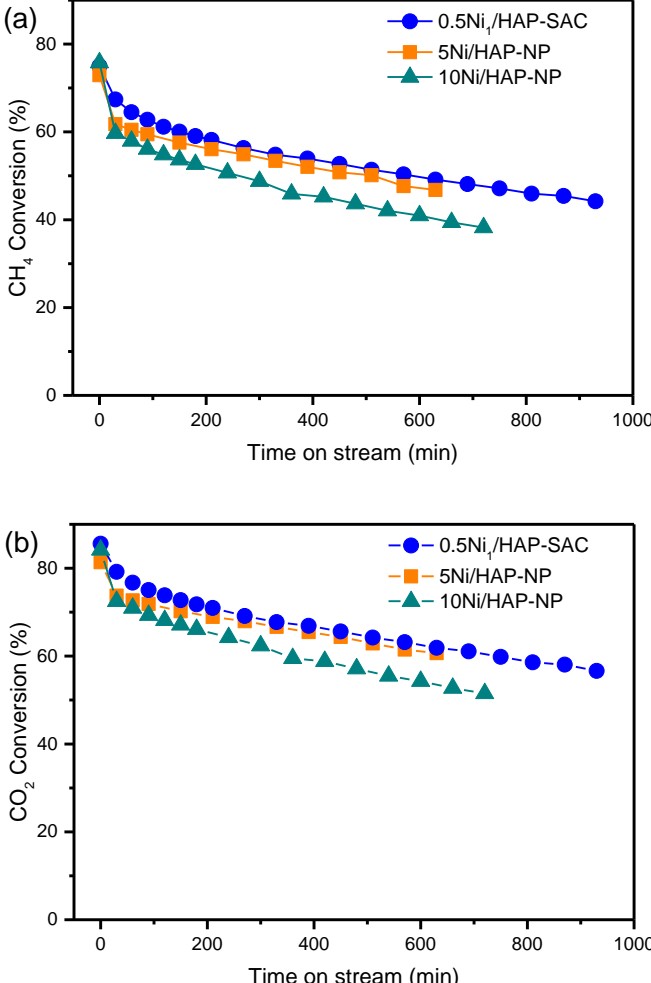

**Figure 7.** (**a**) CH$_4$ conversion during DRM over HAP supported Ni single-atom and nanoparticles catalysts, (**b**) CO$_2$ conversion during DRM over HAP supported Ni single-atoms and nanoparticles catalysts. (All the experiments have been performed at similar metallic gas hourly space velocity: (MGHSV = $12 \times 10^6$ mL $g_{Ni}^{-1}$ h$^{-1}$).

### 3.5.2. Influence of the Synthesis Method of 0.5Ni/HAP

In attempts to reinforce the stability on stream of the prepared single-atom catalysts, the polyvinylpyrrolidone (PVP) assisted synthesis of catalysts was carried out. As shown in Figure 8, with the addition of a small amount of PVP, the initial CH$_4$ (Figure 8a) and CO$_2$ conversion (Figure 8b) reaches 78% and 83%, and only decreases to 68% and 78%, respectively, after 960 min of reaction, exhibiting significantly better stability than the catalyst prepared without surfactant. This enhancement in the catalytic behavior can be attributed to the enrichment of the surface nickel single-atoms on 0.5Ni$_1$/HAP-SAC-PVP, protecting the nickel active site from sintering phenomena during the reaction. In addition, the resulting carbon balance is close to 100% for both catalysts (Figure 8c), proving that carbon deposition is very limited. The H$_2$/CO molar ratio was between 0.68 and 0.97 due to the occurrence of the reverse water gas shift reaction (RWGS).

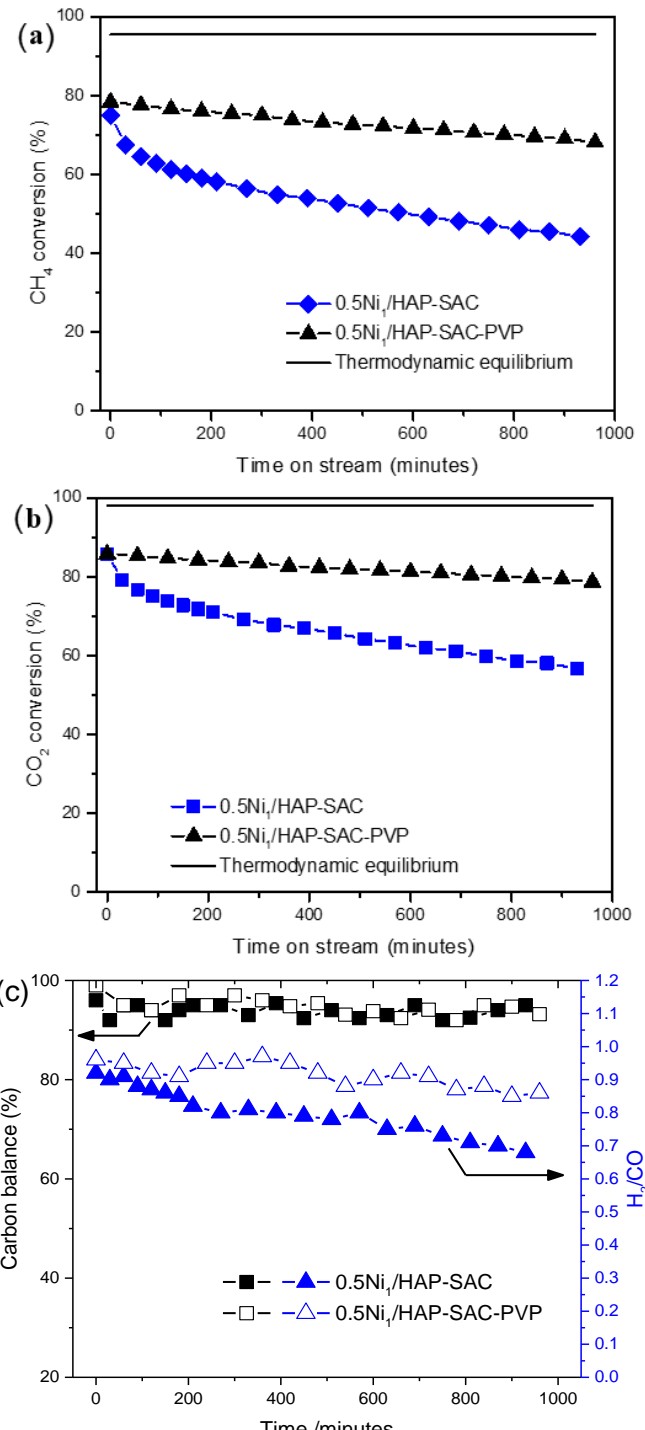

**Figure 8.** (**a**) CH$_4$ Conversion and (**b**) CO2 conversions over 0.5Ni$_1$/HAP-SAC and 0.5Ni$_1$/HAP-SAC-PVP stabilized PVP. (**c**) Carbon Balance and H$_2$/CO ratio over 0.5Ni$_1$/HAP-SAC and 0.5Ni$_1$/HAP-SAC-PVP stabilized PVP.

### 3.5.3. Intrinsic Activity and Comparison with Published Papers

The intrinsic activity of the catalysts was determined at 750 °C and the corresponding TOF (s$^{-1}$) was calculated using a very small amount of catalyst (1.5–2.5 mg) to generate high gas hourly space velocity (GHSV) from 1,056,000 to 1,760,000 mL.g$_{cat}$$^{-1}$.h$^{-1}$ allowing to maintain CH$_4$ and CO$_2$ conversions below 30%. The values of the initial specific reaction rate (mol.g$_{Ni}$$^{-1}$.h$^{-1}$) and the corresponding TOF are listed in Table 1. Nickel single-atom catalyst (0.5Ni$_1$/HAP-SAC) exhibited CH$_4$

and $CO_2$ specific rates of 816.5 and 1186.2 $mol.g_{Ni}^{-1}.h^{-1}$, respectively, and it is four and five times higher than over nickel nanoparticle catalysts (10Ni/HAP-NP). It is important to mention here that $CH_4$ and $CO_2$ corresponding $TOF(s^{-1})$ of nickel single-atoms are almost comparable to that of nickel nanoparticles catalysts.

**Table 1.** Catalytic reaction rate of DRM reaction over different catalysts.

| Catalysts | Ni Loading (wt.%) [a] | Rate (Mol. $g_{Ni}^{-1}.h^{-1}$) | | TOF ($s^{-1}$) | | T (°C) | References |
|---|---|---|---|---|---|---|---|
| | | CH$_4$ | CO$_2$ | CH$_4$ | CO$_2$ | | |
| 0.5Ni$_1$/HAP | 0.51 | 816.5 | 1186.2 | 13.3 | 19.3 | 750 | This work |
| 5Ni/HAP | 4.14 | 339.7 | 414.4 | 11.1 | 14.4 | 750 | This work |
| 10Ni/HAP | 6.98 | 185.2 | 238.6 | 12.1 | 15.5 | 750 | This work |
| Ni/MgO | 7.1 | 16.6 | - | 2.4 | - | 757 | Ref [38] |
| Ni/ZrO$_2$ (3NZH) | 3 | 11.09 | - | 3.32 | - | 700 | Ref [39] |
| Ni/HMS | 6.21 | - | - | 4.98 | | 700 | Ref [40] |
| Ni/ZeO$_2$ | 0.32 | - | - | 29.5 | 37.8 | 800 | Ref [18] |
| Ni/TiO$_2$ | 0.21 | - | - | 12.9 | 18.1 | 800 | Ref [18] |
| Ni/SBA-15-EG | 6.1 | 0.51 | - | 0.06 | - | 450 | Ref [41] |
| LaNiO$_3$ | 23.7 | 1.08 | - | 0.87 | - | 550 | Ref [42] |
| LaNiO$_3$/Al$_2$O$_3$ | 6.9 | 1.9 | - | 1.55 | - | 550 | Ref [42] |
| Ni$_{0.4}$MoC$_x$/SiC | 4.9 | - | - | 28.4 | - | 800 | Ref [43] |

[a] The metal loadings of Ni were detected by ICP-AES.

### 3.5.4. Carbon Deposition and Its Reactivity with $CO_2$ over Nickel Nanoparticle Catalysts

In order to investigate the origin and the reactivity of carbon, two successive experiments were performed. First, $CH_4$ alone was introduced from room temperature to 800 °C after cooling down the reactor, before $CO_2$ was introduced and the temperature was further increased. Using 5Ni/HAP-NP catalyst. The reaction can be expressed as follow:

$$CH_4 \rightarrow 2H_2 + C(s) \tag{2}$$

$$CO_2 + C(s) \rightarrow 2CO \tag{3}$$

Figure 9 shows mass spectrometer (MS) signal of $CH_4$ and $H_2$ on 5Ni/HAP-NP during the experiment. It appears that methane decomposition starts at 430 °C with the simultaneous production of a small amount of $H_2$. The decomposition of $CH_4$ is accelerated by increasing the temperature with a maximum activity obtained at 540 °C according to MS signal (Figure 9a). The subsequent treatment of carbon deposited with $CO_2$ results in the formation of CO only from 647 °C (Figure 9b). These findings are in accordance with those reported recently by Gili et al. where carbon generated from the separate decomposition of both $CH_4$ and CO can be oxidized by $CO_2$ from 600 °C on 5%Ni/MnO [44]. From these experiments, we can conclude that $CH_4$ decomposition on nickel nanoparticles contributes to carbon deposition which can easily react with $CO_2$ to generate CO, suggesting that deposited carbon can be an active intermediate in the DRM reaction over the prepared catalysts.

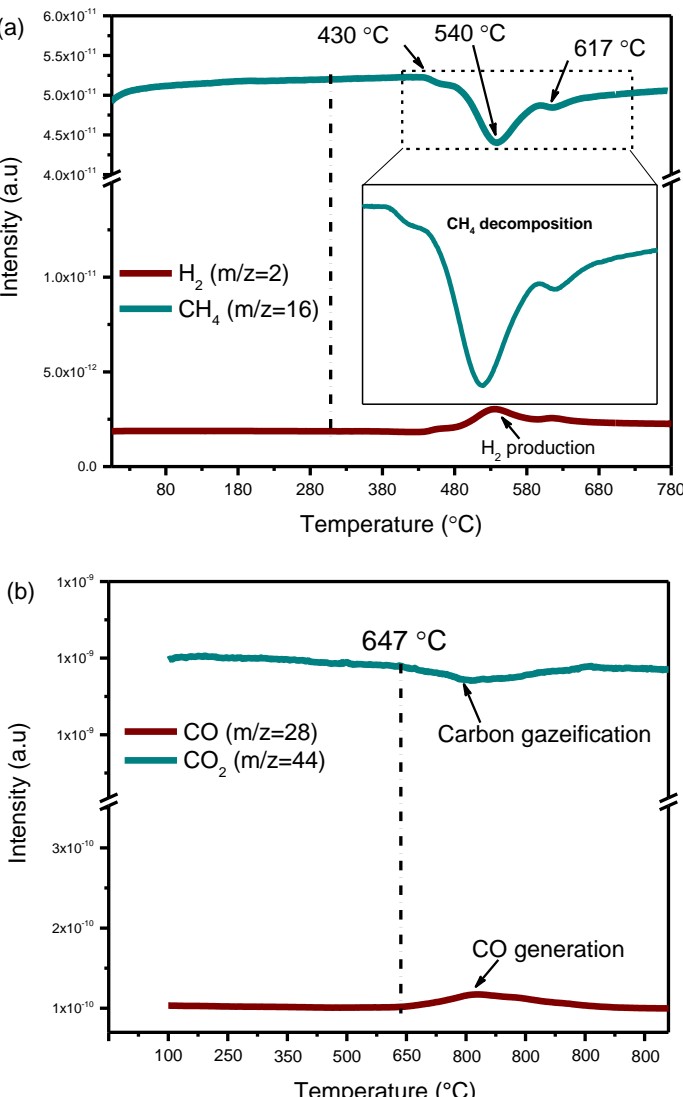

**Figure 9.** MS signal during (**a**) Individual $CH_4$ decomposition; (**b**) Subsequent gasification of deposited carbon with $CO_2$. Conditions: 5Ni/HAP-NP catalyst, T = (25–900 °C). pure $CH_4$ and $CO_2$ were used with total flow of 10 mL $mL^{-1}$.

### *3.6. Characterization after Reaction*

### 3.6.1. TDA-TGA Analysis

The characterization of spent catalysts revealed that the deactivation phenomena can be ascribed to different factors depending on the catalyst. Thermal gravimetric analysis (TGA) exhibited the absence of carbon deposition on nickel single-atoms catalysts (Figure 10). Meanwhile, a high amount of carbon deposition was observed for 5Ni/HAP-NP (4.17% or 1.09 mg.$g^{-1}_{cata}$.$h^{-1}$) and 10Ni/HAP-NP (18.7% or 4.01 mg.$g^{-1}_{cata}$.$h^{-1}$), and the DSC curve showed an exothermic peak for the used 10Ni/HAP-NP catalyst (Figure 10), which fits the temperature zone of coke formation as indicated by TGA.

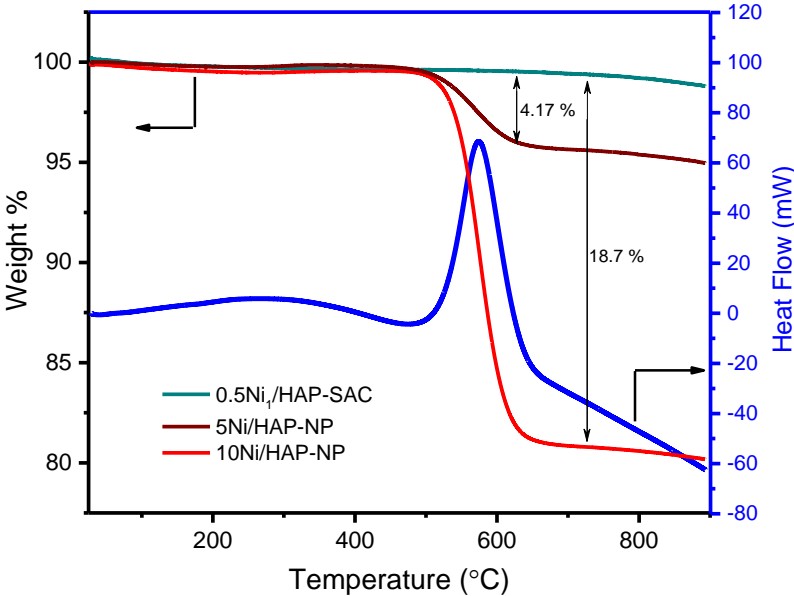

**Figure 10.** TGA profiles and DSC of the spent Ni/HAP nickel catalysts.

### 3.6.2. X-ray Diffraction (XRD)

Figure 11 shows XRD patterns of spent 0.5Ni$_1$/HAP-SAC, 5Ni/HAP as well as 10Ni/HAP samples, whereby the dashed line in the figure designates the position of the carbon diffraction peak. This confirms TGA analysis with the signal of carbon only present for 5Ni/HAP-NP and 10Ni/HAP-NP. The sintering of nickel single atoms is observed after the reaction for 0.5Ni$_1$/HAP-SAC (Figure 12), which can explain the high deactivation rate after a few hours of reaction.

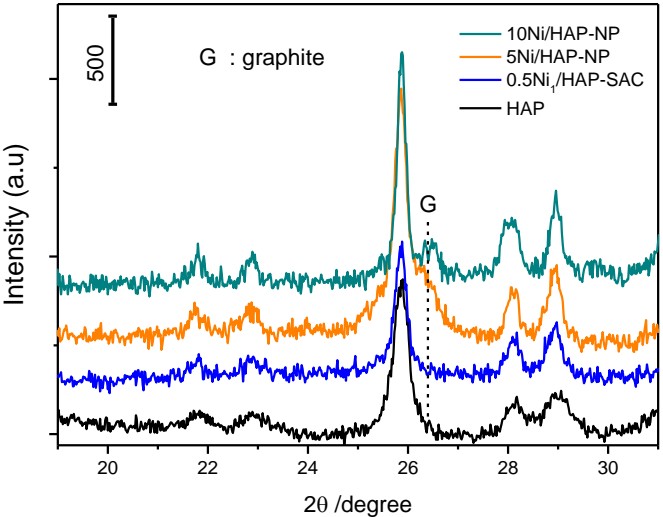

**Figure 11.** XRD patterns of the Ni/HAP catalyst after reaction at 750 °C.

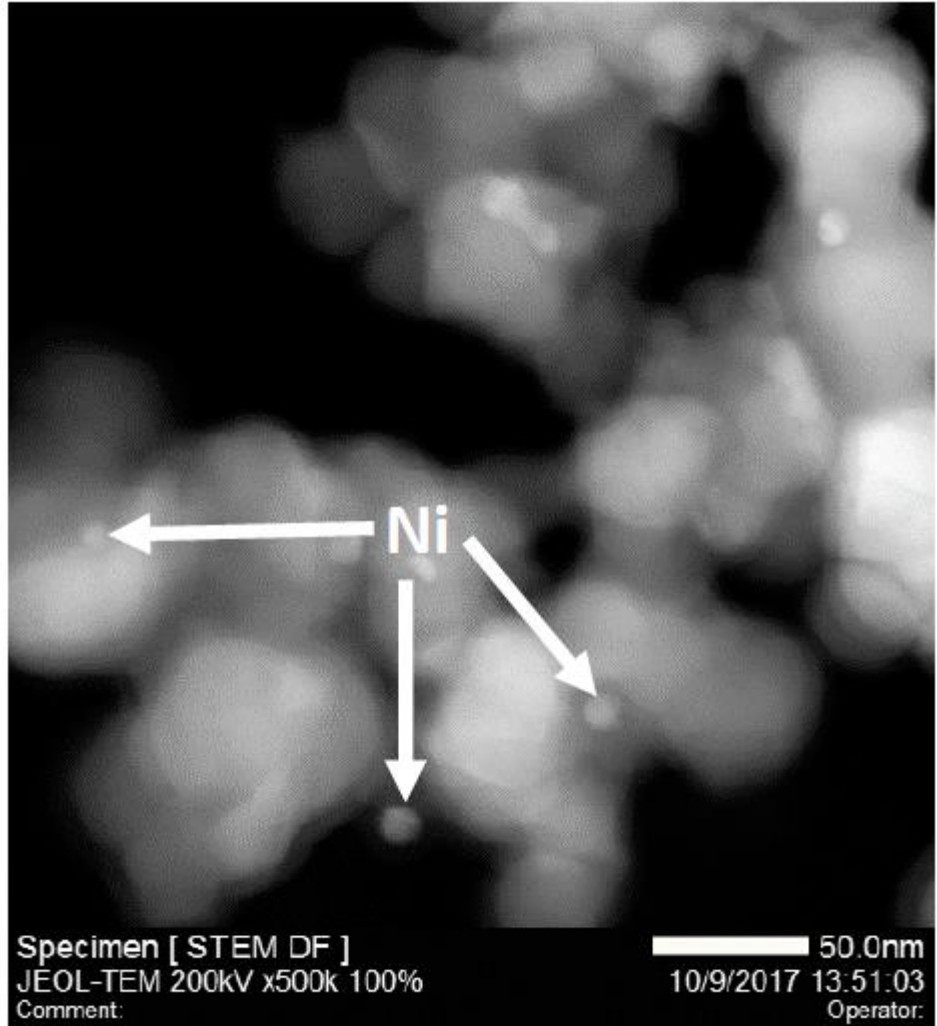

**Figure 12.** STEM image of the 0.5Ni$_1$/HAP-SAC catalyst after reaction at 750 °C.

### 3.6.3. RAMAN Spectroscopy

The carbon deposit was characterized using Raman spectroscopy. As shown in Figure 13, a strong peak at 1328 cm$^{-1}$ (D band) and a peak at 1585 cm$^{-1}$ (G band) are observed for 5Ni/HAP-NP and 10Ni/HAP-NP. These peaks positions are close to the 1346 and 1578 cm$^{-1}$ which have been previously reported in the literature [37]. No peak was observed for 5Ni/HAP-NP, which is in agreement with TGA and XRD results, confirming the absence of carbon deposition on 0.5Ni$_1$/HAP-SAC single atoms catalyst. The graphitization degree was determined by calculating the I$_G$/I$_D$ value which referred to the crystallinity degree, this value is often used as parameter to indicate the crystallinity of the deposited carbon, the higher the ratio (>1) the higher crystallinity [45]. Moreover, 5Ni/HAP-NP and 10Ni/HAP-NP exhibited high and quite similar I$_G$/I$_D$ ratios of 1.22 and 1.19, respectively, which suggests that carbon deposited during DRM at 750 °C is a more ordered graphite structure with a high graphitization nature.

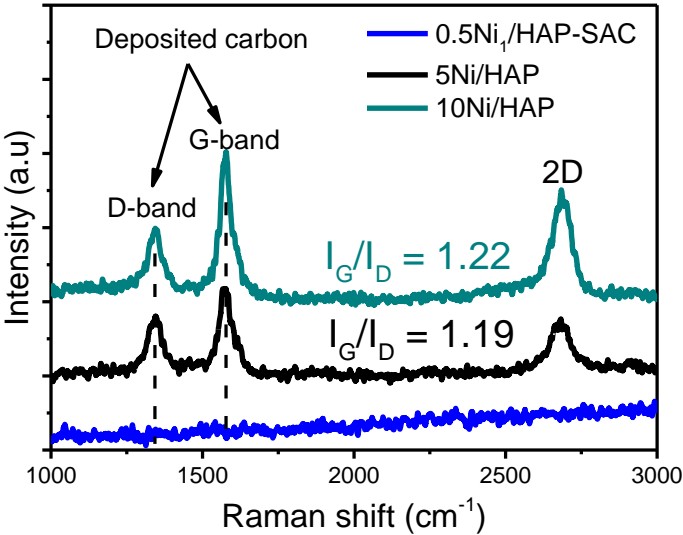

**Figure 13.** Raman spectra of spent $0.5Ni_1$/HAP-SAC and 5Ni/HAP-NP and 10Ni/HAP-NP after reaction at 750 °C.

## 4. Conclusions

In summary, Ni single-atom and nanoparticle catalysts were deposited on hydroxyapatite (HAP) using simple strong electrostatic adsorption (SEA). Ni single-atom catalysts showed the highest reaction rate and excellent resistance to carbon deposition in DRM at 750 °C. The deactivation during time on stream for Ni single-atom catalysts can be attributed mainly to the sintering and aggregation of nickel single-atoms rather than carbon deposit. The use of PVP in the preparation of catalysts leads to a significant improvement in terms of stability. Although Ni nanoparticle catalysts are more resistant to the sintering than Ni single-atom catalysts, a significant amount of carbon deposition is obtained after reaction. It is worth mentioning that carbon deposition can react with $CO_2$ at the temperature of the reaction (750 °C). Thus, such stability of Ni/HAP nanoparticle catalysts in the DRM reaction is more probably related to the different rate of carbon deposition and gasification processes. The present work shows that the use of noble-free, single-atom catalysts is of particular interest for the DRM reaction, and future work must be undertaken to study more deeply its resistance to sintering at high reaction temperature.

**Supplementary Materials:** The following are available online at http://www.mdpi.com/2073-4344/10/6/630/s1, Figure S1: schematic of experimental setup, Figure S2: nitrogen adsorption-desorption isotherms and the corresponding pore size distribution of stoichiometric hydroxyapatite (HAP), Figure S3: STEM image of (a-b) 5Ni/HAP-NP catalysts without reduction, (c) STEM image of 10Ni/HAP-NP catalyst without reduction, Figure S4: STEM image of (a) 5Ni/HAP-NP and (b) 10Ni/HAP-NP catalysts reduced at 500 °C, yellow and red squares indicate respectively some typical nickel nanoclusters and nanoparticles, Figure S5: Dark-field scanning TEM image and EDX elemental mapping of 5Ni/HAP-NP reduced at 500 °C, Figure S6: (a) and (b) STEM images of $0.5Ni_1$/HAP-SAC catalyst without reduction. (c) AC HAADF-STEM image of $0.5Ni_1$/HAP-SAC catalyst without reduction, yellow circles indicate some typical nickel single atoms, Figure S7: $CH_4$ conversion during DRM over fresh and reduced $0.5Ni_1$/HAP catalyst (conditions: T = 550–800 °C, $CH_4/CO_2$/He = 10/10/30, total flow = 50 mL min$^{-1}$, GHSV = 60,000 mL $g_{cat}^{-1}$ h$^{-1}$).

**Author Contributions:** M.A. prepared and characterized the catalysts, and carried out the catalytic tests; A.E.K. and M.A. analyzed the data, contributed to design the experiments and wrote the article; C.B.-D. revised the manuscript; B.Q. acquired financial supports, supervised the work and revised the manuscript. All authors have read and agreed to the published version of the manuscript.

**Funding:** This research was funded by "National Natural Science Foundation of China, grant number 21776270", "National Key Projects for Fundamental Research and Development of China, grant numbers 2016YFA0202801 and 2017YFA0700104", "Strategic Priority Research Program of the Chinese Academy of Sciences, grant number XDB17020100", "DNL Cooperation Fund, CAS, grant number DNL180403" and "LiaoNing Revitalization Talents Program, grant number XLYC1807068".

**Acknowledgments:** The authors acknowledge the performance of synchrotron radiation experiment at the BL14W1 at the Shanghai Synchrotron Radiation Facility, Shanghai Institute of Applied Physics, China.

**Conflicts of Interest:** The authors declare no conflict of interest.

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
