# Peer review of "Highly Active and Carbon-Resistant Nickel Single-Atom Catalysts for Methane Dry Reforming"

_catalysts, doi:10.3390/catal10060630_

Round 1

Reviewer 1 Report

This article is a study of the catalytic properties of nickel single-atom and nanoparticles dispersed on hydroxyapatite. The catalyst is attached to this surface via strong electrostatic absorption. The group was able to characterize the materials generated using X-ray diffraction, STEM, and Raman spectroscopy.

Catalytic activity tests are shown also but fail to clearly present the results obtained. This may be due to the fact that the article is poorly written and quickly exhaust the reader’s attention and comprehension of the goals of the paper.

It seems like this article is a compilation of data from which it is difficult to extract valuable information in an easy way for future readers.

I recommend the authors simplify the article to the main take home points from their study before resubmitting. I will also recommend major editing of the text, especially the introduction.

Author Response

Response to reviewers

Title: Highly active and carbon-resistant nickel single-atom catalysts for methane dry reforming

Response to reviewers

Title: Highly active and carbon-resistant nickel single-atom catalysts for methane dry reforming

Author: Akri et al

Dear Editor and Reviewers,

Thank you very much for your consideration and for the reviewers’ comments concerning our manuscript entitled “Highly active and carbon-resistant nickel single-atom catalysts for methane dry reforming”. Those comments are valuable and very helpful for revising and improving our paper. We have studied comments carefully and have edited and revised all the manuscript which we hope to meet with approval. the responds to the reviewer’s comments are shown below along with the original comments from the reviewers.

Reviewer_1

Comments and Suggestions for Authors

This article is a study of the catalytic properties of nickel single-atom and nanoparticles dispersed on hydroxyapatite. The catalyst is attached to this surface via strong electrostatic absorption. The group was able to characterize the materials generated using X-ray diffraction, STEM, and Raman spectroscopy.

Catalytic activity tests are shown also but fail to clearly present the results obtained. This may be due to the fact that the article is poorly written and quickly exhaust the reader’s attention and comprehension of the goals of the paper.

It seems like this article is a compilation of data from which it is difficult to extract valuable information in an easy way for future readers.

I recommend the authors simplify the article to the main take home points from their study before resubmitting. I will also recommend major editing of the text, especially the introduction.

We thank the reviewer for their recommendations, we are agree with you that the article was poorly written, now we have revised and edited all the manuscript, and we believe is now understandable an pleasant to read.

                                                                                                                                                                                             Action: Manuscript amended

Reviewer 2 Report

The metahne dry reforming reaction to produce syngas is interesting, especially over cheap transition metals when their activity is high and deactivation by sintering or coking is avoided. As a result, this work investigates a relevant topic. However, the results are only briefly mentioned and no discussion is provided. As a result, interpretation is up to the reader. Also, I cannot agree with several important conclusions made by the authors.

For example, the oxidation state of nickel during reaction at 750C is generally metallic, however here the reduction is done at 500 and oxides are still present, which are assumed to exist also during reaction.

The single atom site is in my opinion an exaggeration. There could be a small fraction of Ni single sites present coexisting with nickel crystallites and clusters. A more in-depth analysis, interpretation and discussion of XAS results is urgently required and could clear this issue.

There are many cases of very long sentences (extending well over 3 lines) which makes understanding difficult. Please fragment them.

Also, there are any cases of missing verbs, incorrect sentence structure and adjective/adverb uses. They should be corrected. Please see the highlighed manuscript PDF for details.

The highlighed manuscript PDF also contains my specific questions that need to be addressed

Author Response

Response to reviewers

Title: Highly active and carbon-resistant nickel single-atom catalysts for methane dry reforming

Author: Akri et al

Dear Editor and Reviewers,

Thank you very much for your consideration and for the reviewers’ comments concerning our manuscript entitled “Highly active and carbon-resistant nickel single-atom catalysts for methane dry reforming”. Those comments are valuable and very helpful for revising and improving our paper. We have studied comments carefully and have edited and revised all the manuscript which we hope to meet with approval. the responds to the reviewer’s comments are shown below along with the original comments from the reviewers.

Reviewer_2

Comments and Suggestions for Authors

The methane dry reforming reaction to produce syngas is interesting, especially over cheap transition metals when their activity is high and deactivation by sintering or coking is avoided. As a result, this work investigates a relevant topic. However, the results are only briefly mentioned and no discussion is provided. As a result, interpretation is up to the reader. Also, I cannot agree with several important conclusions made by the authors.

For example, the oxidation state of nickel during reaction at 750C is generally metallic, however here the reduction is done at 500 and oxides are still present, which are assumed to exist also during reaction.

The reviewer raises an interesting question, XPS and TPR show that the chemical state of Ni  in catalysts is consistent with Ni2+ species, with presence of both NiO and Ni(OH)2, in order to confirm the real activate phase, We have conducted previously the experiments with fresh 0.5Ni1/HAP samples without any reduction, the results are presented below, Higher DRM reaction activity was observed through all temperature ranges without reduction strongly evidencing that Ni(OH)x act also as the catalytically active phase. This evidence is highlighted in a revised Figure S7 (see below) and discussed in the manuscript. The performance of the catalysts after reduction at 750 C is more suitable but in current situation due to the epidemic, we need more time to complete this additionally experiments. 

                                                                                                                                                                                    Action: Manuscript+SI amended

The single atom site is in my opinion an exaggeration. There could be a small fraction of Ni single sites present coexisting with nickel crystallites and clusters. A more in-depth analysis, interpretation and discussion of XAS results is urgently required and could clear this issue.

In my best acknowledge, I think AC-STEM image and EXAFS characterizations are enough to confirm such atomically dispersed catalysts. As discussed in our manuscript (line :237-238), EXAFS spectra (Figure 5c bellow) confirmed the dominant presence of Ni-O scattering at 1.5 Å and only a weak Ni-Ni contribution at 2.3 Å. these observations are consistent with atomically dispersed Ni observed by using AC-STEM image (Fig. 5b). This suggests that nickel is not fully dispersed in the atomic scale, but contain also some fraction of Ni nanoclusters which is in agreement with recently reported literature (Nature Communications, 11 (2020) 593). to avoid any confusion, we’ve changed also the title of this work.

                                                                                                                                                                                                    Action: Manuscript amended

There are many cases of very long sentences (extending well over 3 lines) which makes understanding difficult. Please fragment them.

All long sentences in the manuscript is now fragmented and we believe is now more easy for understanding.  

                                                                                                                                                                                          Action: Manuscript amended

Also, there are any cases of missing verbs, incorrect sentence structure and adjective/adverb uses. They should be corrected. Please see the highlighed manuscript PDF for details.

We thank the reviewer for their corrections. missing verbs, incorrect sentence structure and adjective/adverb is now corrected (see new manuscript version).

                                                                                                                                                                                             Action: Manuscript amended

The highlighed manuscript PDF also contains my specific questions that need to be addressed

All relevant specific questions are now addressed (See Manuscript PDF).

                                                                                                                                                                                           Action: Manuscript amended

Reviewer 3 Report

Methane dry reforming is an important process for syngas production. CH4 and CO2 are the most potent greenhouse gases. The rapid catalyst deactivation due to carbon deposition is the challenge, which has been solving by authors in this study.

I have an inconsequential remark:

There is no explanation of PVP abbreviations in the abstract. When we use abbreviations without explanation in the abstract of the article, it makes reading difficult.

Author Response

Response to reviewers

Title: Highly active and carbon-resistant nickel single-atom catalysts for methane dry reforming

Author: Akri et al

Dear Editor and Reviewers,

Thank you very much for your consideration and for the reviewers’ comments concerning our manuscript entitled “Highly active and carbon-resistant nickel single-atom catalysts for methane dry reforming”. Those comments are valuable and very helpful for revising and improving our paper. We have studied comments carefully and have edited and revised all the manuscript which we hope to meet with approval. the responds to the reviewer’s comments are shown below along with the original comments from the reviewers.

Reviewer_3

Comments and Suggestions for Authors

Methane dry reforming is an important process for syngas production. CH4 and CO2 are the most potent greenhouse gases. The rapid catalyst deactivation due to carbon deposition is the challenge, which has been solving by authors in this study.

I have an inconsequential remark:

There is no explanation of PVP abbreviations in the abstract. When we use abbreviations without explanation in the abstract of the article, it makes reading difficult.

We thank the reviewer for this suggestion and have corrected explanation of PVP abbreviations in the abstract.

                                                                                                                                                                                              Action: Manuscript amended

Round 2

Reviewer 1 Report

The first half of the article still contains too many mistakes in english.  I took the liberty to rewrite some sentences... Which shouldn't be the role of the reviewer! Please incorporate and rewrite if need be:

Lines 27-28: Accordingly, the approach used in this publication to control the design of Ni single-atom and nanoparticle catalysts for DRM could enable the development of stable noble-free catalysts.

Lines 42-43: Although noble metals are relatively resistant to carbon deposition compared to non-noble metals,...

Lines 53-54: Hence, the efforts have been focused on the design of nickel catalysts with high resistance to...

Lines 55-58: However, previous work studying the utilization of metal single atom catalysts for DRM reaction are scarcely reported. Only a recent published work describes the synergetic effect between Ni and Ru single atoms on CeO2 support for DRM at low temperature [25].

59-60: In addition to that, the preparation of atomically dispersed single-atom catalysts has recently emerged as a promising method [26-31],...

65-69: Furthermore, hydroxyapatite (HAP) structure was just investigated in a few publications as a catalytic support. HAP has attracted considerable interests due to its ability to make strong metals support interaction [32], in addition to the ability of its structure to undergo different substitutions of both cations (Ca2+ and Ca1+) and anions (PO43– and/or OH–) [33, 34], which can be considered as important sites for anchoring and stabilizing metals single atoms.

73-76: The nickel distribution, its nature and its reactivity were comprehensively investigated by using numerous advanced techniques. Moreover, nickel catalysts based nanoparticles being found favorable sites for carbon deposition. (What does this second sentence even mean?)

185-189: The nickel was not detected at low-loading, which in turn indicates the high dispersion of Ni on the HAP support or formation of very small nanoparticles of Ni on the external surface of HAP that are below the detection limit of XRD. In line with that, the surface's chemical composition was analyzed to check out whether Ni exists at the surface of the catalysts with low-loading.

191: To determine the surface's chemical composition of the catalysts and their chemical states...

Author Response

Response to reviewers

Title: Highly active and carbon-resistant nickel single-atom catalysts for methane dry reforming

Author: Akri et al

Dear Editor and Reviewers,

Thank you very much for your consideration and for the reviewers’ comments concerning our manuscript entitled “Highly active and carbon-resistant nickel single-atom catalysts for methane dry reforming”. Those comments are valuable and very helpful for revising and improving our paper. We have studied comments carefully and have edited and revised the manuscript using Track Changes which we hope to meet with approval. the responds to the reviewer’s comments are shown below along with the original comments from the reviewers.

Comments and Suggestions for Authors

The first half of the article still contains too many mistakes in english.  I took the liberty to rewrite some sentences... Which shouldn't be the role of the reviewer! Please incorporate and rewrite if need be:

Lines 27-28: Accordingly, the approach used in this publication to control the design of Ni single-atom and nanoparticle catalysts for DRM could enable the development of stable noble-free catalysts.

Accordingly, the approach used in this work to design and control the synthesis of Ni single-atom and nanoparticles –based DRM catalysts paves the way towards the development of stable noble metals-free catalysts.

Lines 42-43: Although noble metals-based catalysts are relatively resistant to carbon deposition compared to non-noble metals,...

Although noble metals-based catalysts are relatively more resistant to the undesirable coking compared to non-noble metals catalysts,...

Lines 53-54: Hence, the efforts have been focused on the design of nickel catalysts with high resistance to...

Hence, the efforts were focused on the design of nickel catalysts with high resistance to...

Lines 55-58: However, previous work studying the utilization of metal single atom catalysts for DRM reaction are scarcely reported. Only a recent published work describes the synergetic effect between Ni and Ru single atoms on CeO2 support for DRM at low temperature

Lines 55-58: Recently, the synergy of single-atom sites Ni1 and Ru1 anchored on the CeO2 surface of Ce0.95Ni0.025Ru0.025O2 was evidenced by Tang et al. The authors showed that atoms remain singly dispersed and in a cationic state during catalysis up to 600 °C [25].

59-60: In addition to that, the preparation of atomically dispersed single-atom catalysts has recently emerged as a promising method [26-31],...

Moreover, the preparation of atomically dispersed metal catalysts has recently emerged as a promising method for [26-31],...

65-69: Furthermore, hydroxyapatite (HAP) structure was just investigated in a few publications as a catalytic support. HAP has attracted considerable interests due to its ability to make strong metals support interaction [32], in addition to the ability of its structure to undergo different substitutions of both cations (Ca2+ and Ca1+) and anions (PO43– and/or OH–) [33, 34], which can be considered as important sites for anchoring and stabilizing metals single atoms.

65-69: The Promising behavior of hydroxyapatite (Ca10(PO4)6(OH)2: HAP) supported Ni catalysts in the DRM reaction, was first demonstrated by Boukha et al. [31]. Moreover, SMSI between Au nanoparticles and HAP was evidenced [32]. According to many published papers, varying the Ca/P molar ratio leads to the formation of additional phases and induces significant changes in the textural and acid-base properties [33]. The presence of both cations (Ca2+ and Ca1+) and anions (PO43– and/or OH), can also be considered as important sites for anchoring and stabilizing metals single atoms [34, 35].

73-76: The nickel distribution, its nature and its reactivity were comprehensively investigated by using numerous advanced techniques. Moreover, nickel catalysts based nanoparticles being found favorable sites for carbon deposition. (What does this second sentence even mean?)

This sentence is corrected now, it seems that the editor didn’t send you the last corrected and edited version.

185-189: The nickel was not detected at low-loading, which in turn indicates the high dispersion of Ni on the HAP support or formation of very small nanoparticles of Ni on the external surface of HAP that are below the detection limit of XRD. In line with that, the surface's chemical composition was analyzed to check out whether Ni exists at the surface of the catalysts with low-loading.

185-189: The nickel was not detected at low-loading, which indicates either a high dispersion of Ni on the HAP support or the formation of very small Ni nanoparticles on the external surface of HAP that are below the detection limit of XRD. In order to examine the possibility of having low Ni loading at the surface of the catalyst, we analyzed its chemical composition using…

191: To determine the surface's chemical composition of the catalysts and their chemical states...

191: To determine the chemical composition of the catalysts’ surface and the chemical state of Ni atoms...

Reviewer 2 Report

The questions and suggestions of reviewers are intended to improve the correctness and scientific robustness of this research. As a result, the revised version of manuscript should include the improvements/corrections, as this is the product shown to scientific community. Nobody sees response to reviewers.

If you argue/discuss through comments/annotations and the actual content of revised manuscript is not changed (eventhough it is stated as corrected, see manuscript title of the original and revised version for example), then the revision makes no sense except for wasting someone's time.

Due to no improvements being made and the fact that the corresponding authors who are supposed to be experienced researchers obviously did not check the student's comments and actual revision of paper, I suggest rejection of this work.

Reviewer 3 Report

i am agree with the corrections and do not mind the publication of this article

Round 3

Reviewer 2 Report

I have expressed my decision on not willing to review this particular manuscrpit after the first round of revision.
The authors did nothing of importance (except several cosmetic changes) to improve the manuscript and I suggested rejection. I still stand after my decision.

Author Response

Dear reviewer,

Many thanks for the time given to the review of our work. All your comments and questions are valuable and very helpful for revising and improving the quality of our paper.

I’m agree with you that the edited and revised manuscript with the answers to the questions may not very satisfied because the majority of your questions require the additionally characterization or experiments.

Actually, due to the (Covid 19) epidemic, we are not started yet working in the laboratory and we addressed the questions without recourse to substantive new experiments.

Two others reviewers are satisfied to publish this work. in addition, your decision remains necessary to accept this work.

we kindly ask for the opportunity to change your decision and accept this work for publication in catalysts journal.

Thank you for your understanding
